# Biomedical supervisors' role modeling of open science practices

**Tamarinde L Haven[1]\*, Susan Abunijela[2], Nicole Hildebrand[2]**

[1]Danish Centre for Studies in Research and Research Policy, Department of Political Science, Aarhus University, Aarhus, Denmark; [2]QUEST Center for Responsible Research, Berlin Institute of Health at Charité - Universitätsmedizin Berlin, Berlin, Germany

**Abstract** Supervision is one important way to socialize Ph.D. candidates into open and responsible research. We hypothesized that one should be more likely to identify open science practices (here publishing open access and sharing data) in empirical publications that were part of a Ph.D. thesis when the Ph.D. candidates' supervisors engaged in these practices compared to those whose supervisors did not or less often did. Departing from thesis repositories at four Dutch University Medical centers, we included 211 pairs of supervisors and Ph.D. candidates, resulting in a sample of 2062 publications. We determined open access status using UnpaywallR and Open Data using Oddpub, where we also manually screened publications with potential open data statements. Eighty-three percent of our sample was published openly, and 9% had open data statements. Having a supervisor who published open access more often than the national average was associated with an odds of 1.99 to publish open access. However, this effect became nonsignificant when correcting for institutions. Having a supervisor who shared data was associated with 2.22 (CI:1.19–4.12) times the odds to share data compared to having a supervisor that did not. This odds ratio increased to 4.6 (CI:1.86–11.35) after removing false positives. The prevalence of open data in our sample was comparable to international studies; open access rates were higher. Whilst Ph.D. candidates spearhead initiatives to promote open science, this study adds value by investigating the role of supervisors in promoting open science.

**\*For correspondence:**
tlh@ps.au.dk

**Competing interest:** The authors declare that no competing interests exist.

## Editor's evaluation

This paper will be of interest to scientists who are interested in open-access publishing and open data-sharing procedures. The authors examine associations between PhD candidates' use of open access and open data procedures and use by their supervisors. At present, the study provides solid, useful information suggesting that candidates whose supervisors engage in open-access publishing and open data sharing are more likely to do so, but it does not establish causality or directionality.

## Introduction

When conducted in a manner that emphasizes rigorous and transparent research, supervision can be an important means to socialize Ph.D. candidates into responsible research practices (*Bird, 2001*; *Anderson et al., 2007*; *Davis et al., 2007*; *All European Academies, 2017*; *Universities of The Netherlands, 2018*). Responsible research practices are practices researchers can engage in to

**Table 1.** Prevalence of open access publishing and sharing data openly among unique DOIs.

| Practice | Ph.D. candidates | Supervisors | Total[*] | Spearman's correlation |
|---|---|---|---|---|
| Open access | 548 | 1154 | 1702 (82.8%) | 0.24 |
| Open data (automated) | 67 | 112 | 179 (8.8%) | 0.20 |
| Open data (manually verified) | 34 | 66 | 100 (4.8%) | 0.22 |

[*]Between parentheses indicates proportion out of the total sample.

enhance the transparency, validity, and trustworthiness of their work (**Steneck, 2006**; **Bouter, 2020**). Within biomedical science, examples include open science practices such as openly sharing data and publishing open access, as well as making the underlying methodology openly available, and explicitly acknowledging the limitations of the research findings (**Iqbal et al., 2016**; **Moher et al., 2018**; **Wallach et al., 2018**; **Serghiou et al., 2021**; **Gopalakrishna et al., 2022b**; **Susanin et al., 2022**; **Roche et al., 2022a**; **Hughes et al., 2022**).

To effectively socialize Ph.D. candidates into open and responsible research practices, a pilot study conducted by a research team including the first author distinguished three components (**Haven et al., 2022**). First, the supervisor is supposed to be a role model, i.e., the supervisor engages in open and responsible research practices by for example making their own data and code consistently available. Second, the supervisor encourages the Ph.D. candidate to engage in responsible research practices. After all, it could be that the supervisor has more of a coordinating role or is perhaps versed in another sub-discipline than the Ph.D. candidate's research. Some have referred to this as the distinction between implicit (role modeling) and explicit (verbal instructions and encouragement) supervision (**Fisher et al., 2009**). Third, the supervisor is able to create a psychologically safe atmosphere where Ph.D. candidates feel the space to discuss dilemmas, admit mistakes, and question the status quo (**Antes and DuBois, 2018**; **Antes et al., 2019a**; **Antes et al., 2019b**). This psychological safety in turn contributes to maintaining quality by creating an environment where colleagues can safely scrutinize each others' work (**Roberto et al., 2006**; **Halbesleben and Rathert, 2008**).

Research into responsible supervision is growing, but many knowledge gaps remain. A scoping review (**Pizzolato and Dierickx, 2023**) identified a total of 35 empirical studies on the topic, two-thirds of which used a survey design where they enquired about perceptions from either supervisors or supervisees (except for **Buljan et al., 2018**, who did a qualitative study). More direct evidence (beyond perceptions) on role modeling could not be identified, whereas this role modeling is presumed to be a crucial component of responsible supervision.

This study adds to the literature by proposing a new way to investigate the role modeling of open science practices in biomedicine. It starts from the assumption that if role modeling is important, then it should be possible to discern an association between the supervisor's engagement in open science practices and the Ph.D. candidate's engagement in open science practices. We hypothesized that one should be more likely to identify open science practices (here publishing open access and sharing

**Table 2.** GEE logistic analyses for open access, open data (automated detection), and open data (manually verified).

| Practice | Crude analysis | | | | Adjusted analysis (institution added) | | | |
|---|---|---|---|---|---|---|---|---|
| | N | OR* | 95% CI | p-value | N | OR* | 95% CI | p-value |
| Open access (binary) | 651 | 1.99 | (1.17–3.38) | 0.011 | 651 | 1.64 | (0.94–2.85) | 0.079 |
| Reference category: up to or including the national average (76%) of the supervisor's publications were open access | | | | | | | | |
| Open data automated (binary) | 644 | 2.09 | (1.13–3.88) | 0.019 | 644 | 2.21 | (1.19–4.12) | 0.012 |
| Reference category: supervisor never shared data | | | | | | | | |
| Open data manually verified (binary) | 653 | 3.74 | (1.53–9.12) | 0.004 | 653 | 4.60 | (1.86–11.35) | 0.001 |
| Reference category: supervisor never shared data | | | | | | | | |

N=the total number of included publications by Ph.D. candidates.
*Odds ratios are EXP transformed.

data) in empirical publications that were part of a Ph.D. thesis when the Ph.D. candidates' supervisors engaged in these open science practices compared to those whose supervisors did not or less often did.

## Results

### Open science practices analyses

We managed to include 211 pairs of Ph.D. candidates and supervisors, 50 from Leiden UMC, 54 pairs from Amsterdam UMC, 52 from UMC Groningen, and 55 from Maastricht UMC. This resulted in 2062 DOIs, six of which did not resolve in Unpaywall (0.3%) and 14 PDFs could not be obtained for Oddpub (0.7%). Prevalence for each practice (expressed as unique DOIs) appear in *Table 1*, as well as correlations between the Ph.D. candidates' engagement in a practice and the supervisors' engagement in a practice. GEE logistic regression analyses for both crude and adjusted models appear in *Table 2*.

### Retractions analysis

We were able to link a total of 81,091 publications to the supervisors and Ph.D. candidates. Three Ph.D. candidates could not be identified, all supervisors were identified. Of the 81,091 publications that could be matched to the Ph.D. candidates and supervisors, two were retracted. Both regarded publications where the supervisor appeared as one of the co-authors and were retracted one year after publication. The following reasons were specified, which we interpret as honest errors:

'The authors have become aware that some of the results presented in this paper are invalid, not reproducible, and/or misinterpreted. They consider that the main conclusion of the paper is not valid. They, therefore, retract this publication'.

'The original version of this article was withdrawn by the authors. An error was discovered in the creation of the protein database file that was used for searching, which led to some incorrect associations between peptides and proteins. A corrected version of the manuscript has been supplied which contains very similar peptide identifications as the original, but the resulting number of proteins in various categories has now changed, and as a result, some of the figures and supplementary files have changed also. The underlying conclusions of this study, however, remain unaltered'.

None of the publications included in our own dataset were retracted. RetractionWatch, a blog and database tracking and reporting on retractions (https://retractionwatch.com/), indicated that retractions can take from 3 months to many years, hence some papers may be retracted in the future.

## Discussion

We hypothesized that having a supervisor that shares data or publishes open access was associated with a higher likelihood that the Ph.D. candidate will engage in the same practice. Based on the automated detection of data-sharing statements, we found that having a supervisor that shared data is associated with 2.21 (p=0.012) times the odds to share data when compared to having a supervisor that did not share data. This odds ratio increased to 4.6 (p=0.001) after manually checking the open data statements and removing false positives. The unadjusted open access odds ratio was 1.99 (p=0.011) and became 1.64 (p=0.079) when correcting for the role of the institution. By including the institute variable in our adjusted analyses, the effects of open data remain significant. The odds ratio for the manually verified open data increased by 23%, which could be due to the institution initially masking the effect of the supervisor (see also *Kahan et al., 2014*).

### Contextualisation

Since our sample consists of Ph.D. candidates and supervisors from Dutch UMCs and focuses only on recent years, it may be useful to compare it to international data and reflect on some assumptions that went into our design. *Serghiou et al., 2021* screened all of PubMed Central using the same text-mining algorithm as the current study and found 8.9% of publications to return Open Data statements. It, therefore, seems likely that our sample (8.8%) is not substantially different from the rest of biomedicine.

We included publications across various subfields of biomedicine. However, the odds of finding a publication that shares its underlying data may not be the same for all subfields. Subfields working

with genetics and OMICS data could be more likely to share data than studies that describe clinical research, because of the ethical and privacy-related complications involved (*Mansmann et al., 2023*).

It should be noted that open data does not imply the data is gathered in a rigorous, ethical, and reproducible manner. It could even be that the data are FAIR but still not useful, because the methods used were not well-suited to answer the research question, or because the data collection was sloppy, or because the data fail to capture crucial differences in the target population. We manually verified whether we could find the data, and whether data were actually open, stored on a repository and downloadable. Any assessments about the quality of the data or the quality of archiving (see e.g. *Roche et al., 2022b*) are beyond the scope of this paper.

For open access (82.8% current study), we find our Dutch sample to be in line with national data, but above average when compared to international studies. The Rathenau Institute calculated that 76% of all Dutch publications were available to open access in 2021. Using the Unpaywall, *Robinson-Garcia et al., 2020* found the Dutch uptake of open access to be around 60%. Looking internationally, *Piwowar et al., 2018* assessed the prevalence of open access in different databases. When they looked specifically at biomedicine, using data from the Web of Science, and using the same open/closed distinction as the current study, they found a little over 30% to be open. This difference could be due to various recent Dutch policies for open access. Dutch universities and UMC federations have sealed various deals with publishers (https://www.openaccess.nl/en/in-the-netherlands/publisher-deals), plus the Dutch Research Council requires all their funded research to be published open access (https://www.nwo.nl/en/open-access-publishing).

Open access comes in different forms and publishing in open-access journals is often not for free (*Ross-Hellauer, 2022*). Some publishers make exceptions for scientists from low and middle-income countries, but the Netherlands would not classify. It could thus be that the amount of funding that a supervisor or Ph.D. candidate had available affected the relationship we studied. In other words: funding availability may determine whether Ph.D. candidates (or supervisors) chose to publish in an open access journal. However, it should be noted that green open access, archiving a paper in an appropriate format in an (institutional) repository, can be done free of financial charge.

## The role of early career researchers (ECRs)

A variety of grassroots initiatives that aim to promote open science practices are spearheaded by ECRs (many of them in the process of obtaining a Ph.D.). Popular examples in the Netherlands include ReproducbiliTea (https://reproducibilitea.org) as well as the Open Science Communities (https://www.osc-nl.com).

In addition, many education and training activities to promote open science and responsible research practices target master and Ph.D. candidates. Assuming this group then has more opportunities to learn about open and responsible research, it begs the question of who teaches who. On this note, Pizzolato and Dierickx propose it might be useful to have Ph.D. candidates mentor their supervisors when it comes to matters of research integrity (*Pizzolato and Dierickx, 2022*).

Our findings do not allow for causal inferences, yet we believe they don't need to conflict with ECRs and Ph.D. candidates' knowledge about and engagement in open science practices. Even when one has knowledge about open science practices when starting a Ph.D. trajectory or engages in a ReproducibiliTea reading group during a Ph.D., it may still help to have a supervisor who role models these practices. Considering the associations that we identified, we speculate that working under supervisors who engage in open science themselves could empower Ph.D. candidates to engage in open science more readily. Or at the very least, the supervisor is then less likely to hamper the Ph.D. candidate's engagement in these practices. The other side of the coin, supervisors' lack of engagement in open science practices, still seems more normal, although a recent Dutch survey found Ph.D. candidates to score lower compared to senior researchers on sharing data (*Gopalakrishna et al., 2022a*). Finally, it could be that the relationship investigated here is bidirectional.

## Limitations

This study included many publications by the supervisors, but not all. The number of included first or last author publications for the open science practices varies between 3 and 11; we always included more publications by the supervisor than by the Ph.D. candidate. This meant that at times, we had to exclude pairs because the supervisor did not have a sufficient number of publications, meaning we

may have a small bias towards productive supervisors. In addition, we only included publications up until the year that the Ph.D. candidate defended their thesis, meaning that we at times had to exclude the most recent works.

We only sampled from four out of eight Dutch UMCs; hence our findings may not generalize to all Dutch UMCs, let alone to other countries. That said, we see no prima facie reason to believe that Leiden, Amsterdam - AMC, Groningen, and Maastricht differ substantially from Nijmegen, Utrecht, Rotterdam, and Amsterdam – Vumc, especially given the national data from the Rathenau Institute and the fact that a similar proportion of Open Data statements was returned in a much larger study of biomedical research (*Serghiou et al., 2021*).

Finally, our study does not allow for drawing causal inferences on who educated who regarding open science practices. This is due to its design, but also because we only extracted publications by Ph.D. candidates that were part of their Ph.D. thesis. Hence, we might have missed publications outside the Ph.D. or prior to the Ph.D. candidate that would have indicated a greater engagement in open science practices. That said, this was beyond the scope of our study that aimed at looking at the effect of a supervisor engaging in open science practices.

## Conclusion

We investigated whether having a supervisor that shared data openly and published open access, resulted in a greater odds of the Ph.D. candidate sharing their data and publishing open access. Based on our sample of 211 pairs of biomedical Ph.D. candidates and supervisors, we find the odds of a Ph.D. candidate sharing data to be greater when working under a supervisor who shared data themselves. The effect of open access was smaller and vanished when correcting for institutions, which might be explained by a greater uptake of open access across the Dutch ecosystem. Our design highlights a new way of investigating role modeling in the context of Open Science and other responsible research practices.

## Materials and methods

### Materials availability statement

Data were collected using a pilot-tested protocol that is freely accessible on OSF, we provide a brief overview of our data collection procedures and materials below.

### Ethical aspects

This study used publicly available information (publications) as its data and hence no ethical approval was required. The study was preregistered on the OSF, see: 10.17605/OSF.IO/2PBNS.

### Population

Our population consisted of pairs of Ph.D. candidates and their main supervisors (in the Netherlands, the primary supervisor has to be a full professor, although recently associate professors can get these rights, too). They had to be affiliated with a Dutch University Medical Center (henceforth: UMC) and had to work in biomedicine (understood as their publications being indexed in PubMed).

The Netherlands has eight UMCs, four of those maintained Ph.D. thesis repositories that allowed for the reliable extraction of data (based on a pilot study, see here). These were Leiden UMC, Amsterdam UMC (location AMC), Maastricht UMC, and UMC Groningen, respectively.

### Sample size

In the absence of, to our knowledge, previous studies using a similar method to examine supervisor's role modeling in this or a comparable manner, we conducted a pilot study (n=30). We used the correlations found in the pilot for open access (0.2) as input for the sample size calculation. With an alpha of 0.05 and a power of 0.80, we would need 194 pairs. However, we oversampled as some publications might not meet eligibility criteria after screening the full publication.

### Sampling time

We identified pairs and extracted data between April 17th and June 30th, 2022. We stopped sampling when we passed the required sample size, and wanted to include an equal share of pairs from each

of the four university medical centers. This meant we focused on Ph.D. theses that were defended in 2022 or late 2021.

## Eligibility criteria

Ph.D. candidates' publications had to be in English, part of their Ph.D. thesis (other works published during the same time were excluded), regard empirical work (excluding reviews, commentaries, and narratives), published no earlier than 2018 (to make it reasonable they worked with the supervisor we identified), where the Ph.D. candidate was the sole first author. We only included Ph.D. candidates if they had at least two publications that met these criteria.

Supervisors' publications had to be in English, had to regard empirical work, and be published no earlier than 2017 where the supervisor was the sole first or last author. We only included supervisors if they had at least three publications that met these criteria. Each supervisor only appears once in our dataset to prevent additional clustering and the Ph.D. candidates could not be co-authors on included publications.

## Data extracted

If both the Ph.D. candidate and the supervisor met the eligibility criteria, we extracted the DOIs of the relevant publications, the names of the pairs, the institute they worked at the time of the Ph.D. defense, and the year of the thesis defense. For the supervisor, we also extracted the authorship position.

## Data preparation

To assess the open access status, we used the Unpaywall API through the UnpaywallR package (*Ridel and Franzen, 2022*). The UnpaywallR package takes the DOI and returns the different forms in which a publication is available. We applied the following hierarchy: Gold, Green, Hybrid, Bronze, and Paywalled, following the interpretations of the different forms as described by *Priem, 2021*. We recoded this into a binary variable where Gold, Green, and Hybrid were considered open, and Bronze and paywalled were considered closed.

To identify papers with open data, we used Oddpub followed by a manual review of extracted statements. First, we downloaded the PDFs from the extracted DOIs, we could not access 11 publications – this did not result in excluding pairs. Next, we transformed the PDFs into raw text and applied Oddpub (*Riedel et al., 2020*). Oddpub is a text-mining algorithm that is designed to pick up data-sharing statements in biomedical research papers (*Riedel et al., 2020*) RRID:SCR_018385; Version 6. Publications where Oddpub returned a statement were assigned a one and publications where Oddpub did not return a statement were assigned a zero. We refer to this as open data automated.

To assure that the publications where Oddpub returned a statement genuinely had open data, two extractors manually reviewed all statements using a piloted protocol (*Iarkaeva, 2022*). If there were any discrepancies between extractions, a third extractor or a research data management expert was consulted, and discrepancies were solved through discussion. This resulted in another binary variable where all publications from the list that Oddpub picked up on that had open data were assigned a one and all other publications (i.e. publications where Oddpub initially returned a statement but that were on closer inspection no instances of open data plus all publications where Oddpub returned no statement) were assigned a zero. We refer to this as manually verified open data.

## Data analysis

First, we calculated the prevalence and correlations between a supervisors' engagement in a practice and Ph.D. candidates' engagement in a practice. Next, we used Generalized Equations Estimations (GEE) logistic regression to analyze the data, because our dataset is clustered. We transformed the dataset to the level of the Ph.D. candidate where publications (by the candidate) cluster within the candidate. We recoded supervisors' engagement in open access publishing and data sharing into dichotomous covariates so they could be added to the GEE logistic regression model.

When the percentage of publications from the supervisor that was openly available was above the national average (76%, see *Koens and Vennekens, 2022*), we gave them a 1. If the percentage was 76% or lower, we assigned this supervisor a 0.

We recoded the supervisors' sharing of data into never (no included publications with open data) and ever (one or more included publications with open data) and applied the same categorization to automated statements and manually checked statements. We then exponentially transformed the model's ß coefficients and present Odds ratios. We conducted a crude and adjusted analysis of our odds ratios, where the adjusted models include a dummy coded institute variable to control for a potential confounding bias. In order to determine if the institute was a confounding factor, we compared the measure of association (odds ratios) before and after adjustment. The 10% rule for confounding was applied (*Beukelman and Brunner, 2016*; *Budtz-Jørgensen et al., 2007*).

## Additional retraction analyses

A potential concern with our way of studying supervisors' role modeling regards missing potential irresponsible behaviors. To accommodate this concern, we used the author-disambiguation algorithm developed by *Caron and van Eck, 2014* to obtain meta-data on all publications from supervisors and Ph.D. candidates that were available in the in-house version of Web of Science database at CWTS, Leiden University, the Netherlands, and screened these for retractions. Note that a retraction need not indicate actual irresponsible behavior, it may regard honest mistakes. Where possible, we provide the reason for the retraction as specified by the respective journal.

## Acknowledgements

We would like to acknowledge Martin Holst for his support during the pilot study to assess the feasibility of the approach. Benjamin Gregory Carlisle's support of data science-related issues was crucial. Delwen Franzen helpfully adapted the UnpaywallR script for our project, and Nico Riedel revised the Oddpub script to work on our dataset. Evgeny Bobrov and Anastasiia Iarkaeva instructed us on how to use their protocol and provided the much-needed guidance when assessing challenging cases. Thanks also to Evgeny for pointing out that data sharing may not be equal across all biomedical fields. We also extend our gratitude to Jesper Wiborg Schneider who kindly helped with the retraction analyses and obtaining relevant author publications using the in-house version of the Web of Science database at CWTS, Leiden University, the Netherlands.

## Additional information

### Funding

| Funder | Grant reference number | Author |
| --- | --- | --- |
| Nederlandse Organisatie voor Wetenschappelijk Onderzoek | 019.212SG.022. | Tamarinde L Haven |

The funders had no role in study design, data collection and interpretation, or the decision to submit the work for publication.

### Author contributions

Tamarinde L Haven, Conceptualization, Formal analysis, Supervision, Investigation, Methodology, Writing – original draft, Project administration, Writing – review and editing; Susan Abunijela, Nicole Hildebrand, Data curation, Formal analysis, Investigation, Project administration, Writing – review and editing

### Author ORCIDs

Tamarinde L Haven http://orcid.org/0000-0002-4702-2472

### Decision letter and Author response
Decision letter https://doi.org/10.7554/eLife.83484.sa1
Author response https://doi.org/10.7554/eLife.83484.sa2

# Additional files

## Supplementary files
• MDAR checklist

## Data availability
All data are available alongside the code to produce it on the associated GitHub repository (copy archived at *Haven et al., 2023*).

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
