## [Editor Report]

This paper will be of interest to scientists who are interested in open-access publishing and open data-sharing procedures. The authors examine associations between PhD candidates' use of open access and open data procedures and use by their supervisors. At present, the study provides solid, useful information suggesting that candidates whose supervisors engage in open-access publishing and open data sharing are more likely to do so, but it does not establish causality or directionality.

---

## [Decision Letter]

**Decision letter after peer review:**

Thank you for submitting your article "Meta-Research: Biomedical supervisors' role modeling of responsible research practices" for consideration by *eLife*. Your article has been reviewed by 2 peer reviewers, and the evaluation has been overseen by a Reviewing Editor and Mone Zaidi as the Senior Editor. The following individuals involved in the review of your submission have agreed to reveal their identity: Lisa Schwiebert (Reviewer #1); Jon Agley (Reviewer #2).

Essential revisions:

The investigators are encouraged to widen the years of their analyses so as to include the number and incidence of retracted papers by respective supervisor and PhD supervisee pairs.

Please also address Reviewer 2's recommendations for the authors.

*Reviewer #1:*

The purpose of the current study is to address a gap in knowledge regarding the assessment of responsible supervision of PhD supervisees in the field of biomedical research. Specifically, the investigators theorize that a supervisor's role modeling of responsible research practices in the context of data sharing will likely result in the PhD supervisees engaging in the same responsible practice as compared with supervisors who did not data share. Through careful analyses of Open Access publishing and Open Data sharing platforms, the investigators found that the odds of a PhD supervisee sharing data were greater working with a supervisor who shared data themselves versus those who did not.

The strengths of this study are several and they include an innovative approach toward a complex concern; strong statistical analyses; well-described limitations of the study. Overall, this is an interesting report, while not wholly surprising, it does add value with its evidenced-based approach toward the assessment of responsible research practices.

Of note, the conclusion assumes that shared data themselves are accurate, rigorous, and reproducible, which may not be wholly representative of the desired responsible practices.

The investigators are encouraged to widen the years of their analyses so as to include the number and incidence of retracted papers by respective supervisor and PhD supervisee pairs.

*Reviewer #2:*

The authors sought to undertake an examination of open-access publishing and open data sharing among PhD students and their supervisors as part of an expressed interest in responsible research practices (RRPs). The study was preregistered (including hypotheses, procedures, and outcomes), contained a step-by-step guide with screenshots for replicating their protocol, and shared all data and code.

The study results are fairly clearly explained, though I have some specific comments and questions that I provide to the authors privately. The research question itself is interesting and the procedures used to identify and match open access publication and data availability both use and advance novel procedures for doing so. I do have some questions about possible mediating and moderating factors that may be important to discuss, as well as the possible importance of separating open publication and data accessibility from the highly related, but not identical, concept of RRP. In most cases, the resolution of these questions would not change the core findings of the manuscript but would simply serve to advance clarity and discussion. As with all papers using associative analyses, readers are cautioned to avoid causal interpretations – a caveat that is also addressed by the paper itself.

Abstract

Although I have a few questions about some of the analyses, if indeed it is the case that the open access odds were lower, since your manuscript has 4 primary findings, consider sharing all 4 in the abstract.

Introduction

Paragraph 1: The first sentence appears to imply that supervision generally results in socialization into responsible research practices. This may benefit from being expanded somewhat for clarity. For example, the first paper (Bird, 2001) discusses how role modeling and mentorship differ in key ways, and the second paper (Anderson et al., 2007) explicitly notes that certain kinds of mentorship/supervision were associated with higher rates of problematic practices for early career researchers. Would it be more apt to state something like, "When conducted in a manner that clearly emphasizes rigorous, reproducible, and transparent research practices, it is plausible that supervision…"

Paragraph 1: The sentence focused on examples of RRPs cites articles that address RRPs to a degree, but some (e.g., Gopalakrishna et al., 2022) primarily focus on questionable research practices, not necessarily why RRPs might mitigate some of those concerns. I encourage the identification of additional references on RRP to be included here.

Paragraph 2: Although scientific writing conventions vary, my own preference would be for your description to be in the personal possessive here (e.g., "…a pilot study conducted by the first author identified three components."). I think it is important for the reader to understand the origin of these ideas and that you are building on your own prior work.

Paragraph 2: In the final sentence, consider also adding a point raised by the cited work around the importance of being able to safely scrutinize each others' work.

Materials and methods

Materials Availability: I appreciated the excellent use of screenshots to guide readers through the procedures used (as documented on OSF).

Population: I encourage including some of the population information from your preregistration here as well. For example, readers may not intuit that full professorship is the normative rank for candidate supervision in Dutch universities and might mistakenly assume some sort of selection bias (e.g., if associate professors were more commonly supervisors but you identified mostly full professors…).

Sampling Time: It is unclear why the data extraction timeframe resulted in the specific range of PhD thesis defense times. You explore some of this in your preregistration, but additional details would be useful here (even if it's just to indicate that normative delays reflected X – this also addresses why you planned to but did not include German universities, in part).

Eligibility Criteria: Can you clarify whether "latest 2018" means "no earlier than 2018"? I suspect this is a linguistic difference but I want to be sure. In addition, can you explain why you included PhD candidates if they only had at least 2 publications? Is it possible that this inclusion criterion systematically excluded candidates or supervisors with specific characteristics?

Data Preparation: I recommend referencing the Unpaywall FAQ here for data definitions (can be cited as a resource since the FAQ was prepared by Jason Priem; https://support.unpaywall.org/support/solutions/articles/44001777288-what-do-the-types-of-oa-status-green-gold-hybrid-and-bronze-mean-). The advantage is that, as the FAQ notes, there is no formal classification schema that exists. I would not have known that "Green" OA includes institutional repositories because some journals use that term similarly to how this FAQ uses "Hybrid."

Data Analysis: I am not a statistician, but I wonder (perhaps naively) whether 'university' should have been included in the model as a clustering term, especially since some of the earlier literature cited indicates that the environment (beyond the mentor and lab) can contribute to RRP.

Discussion

The first paragraph of the Discussion contains information that I would ordinarily associate with Results. That aside, can you clarify the interpretation of the open-access information? If these are presented as exp(b) values, then would a.18 would indicate that PhD candidates published open access less often when their supervisors did than when they did not?

Contextualization: Since there is a substantial cost associated with publishing in many open-access venues, might an important contributor to variance (that might affect this model but is not included in the model) be the funding amount of the supervisor? For example, some supervisors may have limited ability to publish open access where a cost is incurred, some may be able to publish their own work (or at least some of their own work) open access, but fewer likely can afford to support their candidates in publishing at cost. If there is indeed a weaker relationship between supervisor publishing and candidate publishing than between data availability, then could this be a mediating or moderating factor?

The role of ECRs: I'm not sure that we should assume that ECRs have more opportunities to learn about responsible research, in general, though they may have more exposure to open research principles. This paper primarily focuses on open data and open publication. While these are important components of reproducibility and transparency, and while they may aid in the identification of problematic findings and work, they do not subsume the entirety of RRPs. Some components of RRP are longstanding or axiomatic ethical determinations or orientations, whereas the rise of open access has been relatively recent. So even highly ethical, responsible, and transparent senior researchers may be slow to uptake new publishing approaches.

The role of ECRs: Since your findings are nondirectional (e.g., correlations), it seems plausible (given the citations you provide) that you may be capturing a bidirectional relationship.

[Editors' note: further revisions were suggested prior to acceptance, as described below.]

Thank you for resubmitting your work entitled "Biomedical supervisors' role modeling of open science practices" for further consideration by *eLife*. Your revised article has been evaluated by Mone Zaidi (Senior Editor) and a Reviewing Editor.

The manuscript has been improved but there are some remaining issues to consider and respond to from Reviewer #2:

*Reviewer #1 (Recommendations for the authors):*

The authors have well responded to the initial review – no further comments or concerns.

*Reviewer #2 (Recommendations for the authors):*

I would like to thank the authors for the work put into this revision and the clarifications offered in the response to reviewers. I restate my perception that this work is interesting and provides a solid and useful contribution to the overall scientific enterprise in this area. I have a few questions and comments around the revised work that I hope you find useful.

Abstract

Here and elsewhere, you might want to use or provide the specific meaning of "actively published open access" since it's very specific to this article (e.g., published open access more often than the average Dutch professor did in 2021). There are also comments in the Discussion about the odds ratio that apply to the abstract (depending on whether you decide to make those revisions).

Introduction

I think that the modifications made to the Introduction section are helpful and address my comments in full.

Materials and methods

I am glad that my questions prompted further investigation into the analyses and variable structure. My only remaining comment relates to the new statement, "The 10% rule for confounding was applied." I think the application here is fine (again – I am not a professional statistician, so caveat emptor) but a citation may be useful. Some readers may be unfamiliar with that rule, and others may have questions about it (see, eg., Lee, 2014 – https://doi.org/10.2188%2Fjea.JE20130062).

Results

In the title for Table 1, you might want to specify the unit of analysis (e.g., a unique DOI). While it is clear to me what is meant by the table, a reader who is skimming might assume that there are, for example 548 PhD candidates publishing open access, rather than 548 DOIs from PhD candidates that were published open access.

If I understand your methods for identifying open data correctly, it may be worth removing the "Open data (automated)" information from the Results entirely. Here is my reasoning: Oddpub was used to extract possible instances of open data statements, but through manual verification, you triaged around half of those instances. Specifically, it seems that those that were excluded by manual verification did not have open data statements as affirmed by two to three human reviewers. Thus, I am not sure what value is added by analyzing the "Open data (automated)" variable, which seems to only describe a subset of papers for which a specific algorithm thought it found open data. We are not concerned with whether things are associated with whether a machine thinks there might be an open data statement, but rather whether things are associated with actual open data access.

Can you please consider verifying the total number of linked publications (81,091)? Given that the supervisor+candidate total n was around 2,000, that number would imply that the supervisors in this study published an additional 79,000 or so papers before 2017. While that is certainly possible, even if we assume that the 211 recent PhD candidates published an aggregate 10,000 papers (a very generous assumption), it would mean that the 211 supervisors published an average of 327 papers each, which is fairly remarkable. In some ways, this number is less relevant than the 2 retractions, but it does make me wonder.

Discussion

If you agree with my point about automated open data statement identification, you can also remove this sentence from the Discussion ("Based on the automated detection of data sharing statements, we found that having a supervisor that shared data is associated with 2.21 times the odds to share data when compared to having a supervisor that did not share data.")

I would suggest revising this sentence: "Effects for Open Access were smaller (1.99) and became nonsignificant when correcting for the role of the institution." Instead, you could indicate that the unadjusted odds ratio was 1.99 (p=.011) and the adjusted odds ratio was 1.64 (p=.079). I think that the shift in odds is interesting even though it falls above the conventional significance threshold.

You currently write, "This may indicate there being greater acceptance of and support for a particular practice in a research group. In other words: a responsible supervisory climate may empower PhD candidates to engage in open science more readily." You might want to contextualize this with something like, "In considering the associations that we identified, we speculate that…"

---

## [Author Response]

Essential revisions:The investigators are encouraged to widen the years of their analyses so as to include the number and incidence of retracted papers by respective supervisor and PhD supervisee pairs.Please also address Reviewer 2's recommendations for the authors.

Thank you for these valuable suggestions. We have widened the years of analyses to assess retractions (described in more detail below) and incorporated the recommendations by Reviewer 2 to the best of our ability.

Reviewer #1:The purpose of the current study is to address a gap in knowledge regarding the assessment of responsible supervision of PhD supervisees in the field of biomedical research. Specifically, the investigators theorize that a supervisor's role modeling of responsible research practices in the context of data sharing will likely result in the PhD supervisees engaging in the same responsible practice as compared with supervisors who did not data share. Through careful analyses of Open Access publishing and Open Data sharing platforms, the investigators found that the odds of a PhD supervisee sharing data were greater working with a supervisor who shared data themselves versus those who did not.The strengths of this study are several and they include an innovative approach toward a complex concern; strong statistical analyses; well-described limitations of the study. Overall, this is an interesting report, while not wholly surprising, it does add value with its evidenced-based approach toward the assessment of responsible research practices.Of note, the conclusion assumes that shared data themselves are accurate, rigorous, and reproducible, which may not be wholly representative of the desired responsible practices.

We agree and added this point to the Contextualisation section (part of Discussion) noting open data does not imply ethically, sound and rigorously collected data, see below. We also adapted our terminology and now use Open Science Practices throughout the paper.

“It should be noted that open data does not imply the data is gathered in a rigorous, ethical, and reproducible manner. It could even be that the data are FAIR but still not useful, because the methods used were not well-suited to answer the research question, or because the data collection was sloppy, or because the data fail to capture crucial differences in the target population. We manually verified whether we could find the data, whether data were actually open, stored on a repository and downloadable.Any assessments about the quality of the data or the quality of archiving (see e.g., Roche et al., 2022) are beyond the scope of this paper.”

The investigators are encouraged to widen the years of their analyses so as to include the number and incidence of retracted papers by respective supervisor and PhD supervisee pairs.

Thank you for this interesting suggestion. We used a modified approach to answer this request. First we collected additional data on the supervisor and supervisee pairs, namely their email addresses and their Open Researcher and Contributor IDs (ORCIDs).

Second we used our existing data plus the newly acquired information to identify the authors and their papers using the author-disambiguation algorithm developed by Caron & van Eck (2014) in the in-house version of Web of Science database at CWTS, Leiden University, the Netherlands.

Web of Science data includes a binary variable on retractions. Note that the time window for retractions tends to be rather long (see e.g., https://retractionwatch.com/2017/07/07/retraction-countdown-quickly-journals-pull-papers/). Hence it remains possible that papers we included may be retracted in the future.

Using this approach, we identified a total of 81091 publications where the researchers in our sample were identified as (co-)authors. We found 2 retracted papers. Both regarded publications where the supervisor appeared as co-author and both regarded honest mistakes where the retraction was issued one year after the initial publication. We include the specifications provided by the journal below:

Original paper: British Journal of Pharmacology (2002) 136, 1107–1116. Doi:10.1038/sj.bjp.0704814

Retraction: British Journal of Pharmacology (2003) 138, 531. Doi:10.1038/sj.bjp.0705183

Retraction notice: The authors have become aware that some of the results presented in this paper are invalid, not reproducible and/or misinterpreted. They consider that the main conclusion of the paper is not valid. They therefore retract this publication.

Original paper: Molecular & Cellular Proteomics (2018) 17, 2132-2145. Doi: 10.1074/mcp.RA118.000792

Retraction: Molecular & Cellular Proteomics (2019) 16, 1270. Doi:10.1074/mcp.W119.001571

Retraction notice: The original version of this article was withdrawn by the authors. An error was discovered in the creation of the protein database file that was used for searching, which led to some incorrect associations between peptides and proteins. A corrected version of the manuscript has been supplied which contains the very similar peptide identifications as the original, but the resulting number of proteins in various categories has now changed, and as a result some of the figures and supplementary files have changed also. The underlying conclusions of this study, however, remain unaltered.

Corrected version: Shraibman, B., Barnea, E., Kadosh, D. M., Haimovich, Y., Slobodin, G., Rosner, I., López-Larrea, C., Hilf, N., Kuttruff, S., Song, C., Britten, C., Castle, J., Kreiter, S., Frenzel, K., Tatagiba, M., Tabatabai, G., Dietrich, P.-Y., Dutoit, V., Wick, W., Platten, M., Winkler, F., von Deimling, A., Kroep, J., Sahuquillo, J., Martinez-Ricarte, F., Rodon, J., Lassen, U., Ottensmeier, C., van der Burg, S. H., Thor Straten, P., Poulsen, H. S., Ponsati, B., Okada, H., Rammensee, H. G., Sahin, U., Singh, H., and Admon, A. (2019) Identification of tumor antigens among the HLA peptidomes of glioblastoma tumors and plasma. Mol. Cell. Proteomics 18, 1255–1268.

In case the reviewer would like to see these findings in the paper, we have prepared the following text. That said, we feel that given the small number of retractions (#2) and the provided reasons (honest mistakes), it may not add much to the paper. We leave it up to the reviewer and the editor to decide on the matter and have attached the data underlying in anonymised form to this re-submission.

“Additional analyses

A potential concern with our way of studying supervisors’ role modeling regards missing potential irresponsible behaviors. To accommodate this concern, we used the author-disambiguation algorithm developed by Caron & van Eck (2014) to obtain meta-data on all publications from supervisors and PhD candidates that were available in the in-house version of Web of Science database at CWTS, Leiden University, the Netherlands, and screened these for retractions. Note that a retraction need not indicate actual irresponsible behavior, it may regard honest mistakes. Where possible, we provide the reason for the retraction as specified by the respective journal.

…

Retractions analysis

We were able to link a total of 81091 publications to the supervisors and PhD candidates. Three PhD candidates could not be identified, all supervisors were identified. Of the 81091 publications that could be matched to the PhD candidates and supervisors, 2 were retracted. Both regarded publications where the supervisor appeared as one of the co-authors and were retracted one year after publication. The following reasons were specified, which we interpret as honest errors:

“The authors have become aware that some of the results presented in this paper are invalid, not reproducible and/or misinterpreted. They consider that the main conclusion of the paper is not valid. They therefore retract this publication.”

“The original version of this article was withdrawn by the authors. An error was discovered in the creation of the protein database file that was used for searching, which led to some incorrect associations between peptides and proteins. A corrected version of the manuscript has been supplied which contains the very similar peptide identifications as the original, but the resulting number of proteins in various categories has now changed, and as a result some of the figures and supplementary files have changed also. The underlying conclusions of this study, however, remain unaltered.”

None of the publications included in our own dataset were retracted. RetractionWatch, a blog and database tracking and reporting on retractions (https://retractionwatch.com/), indicated that retractions can take from 3 months to many years, hence some papers may

be retracted in the future.”

Reviewer #2:The authors sought to undertake an examination of open-access publishing and open data sharing among PhD students and their supervisors as part of an expressed interest in responsible research practices (RRPs). The study was preregistered (including hypotheses, procedures, and outcomes), contained a step-by-step guide with screenshots for replicating their protocol, and shared all data and code.The study results are fairly clearly explained, though I have some specific comments and questions that I provide to the authors privately. The research question itself is interesting and the procedures used to identify and match open access publication and data availability both use and advance novel procedures for doing so. I do have some questions about possible mediating and moderating factors that may be important to discuss, as well as the possible importance of separating open publication and data accessibility from the highly related, but not identical, concept of RRP. In most cases, the resolution of these questions would not change the core findings of the manuscript but would simply serve to advance clarity and discussion. As with all papers using associative analyses, readers are cautioned to avoid causal interpretations – a caveat that is also addressed by the paper itself.

Thank you for the kind words and helpful comments. We re-ran our analyses with institution as a potential confounding variable and present the crude and adjusted results side-by-side in Table 2 now. We have also adjusted the formulation of our concept, and the paper now consistently refers to open science practices. The remainder of the comments is answered in-depth below.

AbstractAlthough I have a few questions about some of the analyses, if indeed it is the case that the open access odds were lower, since your manuscript has 4 primary findings, consider sharing all 4 in the abstract.

Thanks for flagging this, we added the open access odds to the abstract. We added the following sentence:

“Having a supervisor who actively published open access was associated with an odds of 1.99 to publish open access, but this effect became nonsignificant when correcting for institutions.”

IntroductionParagraph 1: The first sentence appears to imply that supervision generally results in socialization into responsible research practices. This may benefit from being expanded somewhat for clarity. For example, the first paper (Bird, 2001) discusses how role modeling and mentorship differ in key ways, and the second paper (Anderson et al., 2007) explicitly notes that certain kinds of mentorship/supervision were associated with higher rates of problematic practices for early career researchers. Would it be more apt to state something like, "When conducted in a manner that clearly emphasizes rigorous, reproducible, and transparent research practices, it is plausible that supervision…"

That would indeed be more apt – we revised the sentence to emphasize that a particular style of supervision is necessary, namely responsible supervision. We now write:

“When conducted in a manner that emphasizes rigorous and transparent research, supervision can be an important manner to socialize PhD candidates into responsible research practices.”

Paragraph 1: The sentence focused on examples of RRPs cites articles that address RRPs to a degree, but some (e.g., Gopalakrishna et al., 2022) primarily focus on questionable research practices, not necessarily why RRPs might mitigate some of those concerns. I encourage the identification of additional references on RRP to be included here.

Thank you for flagging this, we added an additional 6 references that investigate outcome variables similar to ours, i.e., Iqbal et al. (2016); Wallach et al. (2018); Susanin et al. (2022); Roche et al. (2022), and Hughes et al. (2022).

Paragraph 2: Although scientific writing conventions vary, my own preference would be for your description to be in the personal possessive here (e.g., "…a pilot study conducted by the first author identified three components."). I think it is important for the reader to understand the origin of these ideas and that you are building on your own prior work.

It was indeed out of different conventions that we choose this description. We now rephrased it to: “… a pilot study conducted by a research team including the first author identified three components.” to denote the connection with previous research.

Paragraph 2: In the final sentence, consider also adding a point raised by the cited work around the importance of being able to safely scrutinize each others' work.

Thank you for this suggestion, we added the following point with references:

“This psychological safety in turn contributes to maintaining quality by creating an environment where colleagues can safely scrutinize each others’ work (Roberto, Bohmer, and Edmondson, 2006; Halbesleben & Rathert, 2008).”

Materials and methodsMaterials Availability: I appreciated the excellent use of screenshots to guide readers through the procedures used (as documented on OSF).

We are pleased to read you found it useful.

Population: I encourage including some of the population information from your preregistration here as well. For example, readers may not intuit that full professorship is the normative rank for candidate supervision in Dutch universities and might mistakenly assume some sort of selection bias (e.g., if associate professors were more commonly supervisors but you identified mostly full professors…).

This is very helpful, especially given the international readership. We added the following clarifier:

“(in The Netherlands, the primary supervisor has to be a full professor, although recently associate professors can get these rights, too).”

Sampling Time: It is unclear why the data extraction timeframe resulted in the specific range of PhD thesis defense times. You explore some of this in your preregistration, but additional details would be useful here (even if it's just to indicate that normative delays reflected X – this also addresses why you planned to but did not include German universities, in part).

The sampling time frame of 2022-2021 was a result of the required sample size. We set out to include about 200 pairs and wanted to assure equal representation among the four institutions. For some institutions, this meant that we had to go back until theses defended late 2021. We now added the following clarification:

“We stopped sampling when we passed the required sample size, and wanted to include an equal share of pairs from each of the four university medical centers.”

Eligibility Criteria: Can you clarify whether "latest 2018" means "no earlier than 2018"? I suspect this is a linguistic difference but I want to be sure. In addition, can you explain why you included PhD candidates if they only had at least 2 publications? Is it possible that this inclusion criterion systematically excluded candidates or supervisors with specific characteristics?

Indeed, no earlier than 2018. This has been adjusted in the main text now. Initially, the criterion of at least 2 publications was intended to filter our German thesis that were Dr Med instead of PhD degrees – Dr Med theses can be built around one journal publication. In addition, Dutch university medical centers often apply the criterion of at least 2 journal publications the regulations of Maastricht University and Amsterdam University Medical center even denote this more explicitly. Hence, we see no reason to believe this systematically excluded PhD candidates. That said, we encountered cases where we could not include a pair because the supervisor did not have enough works, either as primary or last author, or works without the PhD candidate as a co-author. We reflect on this on the Limitations’ section as follows:

“This meant that at times, we had to exclude some pairs because the supervisor did not have a sufficient number of publications, meaning we may have a small bias towards productive supervisors.”

Data Preparation: I recommend referencing the Unpaywall FAQ here for data definitions (can be cited as a resource since the FAQ was prepared by Jason Priem; https://support.unpaywall.org/support/solutions/articles/44001777288-what-do-the-types-of-oa-status-green-gold-hybrid-and-bronze-mean-). The advantage is that, as the FAQ notes, there is no formal classification schema that exists. I would not have known that "Green" OA includes institutional repositories because some journals use that term similarly to how this FAQ uses "Hybrid."

Excellent suggestion, we added the following sentence:

“We applied the following hierarchy: Gold, Green, Hybrid, Bronze, and Paywalled, following the interpretations of the different forms as described by Priem (2021).”

Data Analysis: I am not a statistician, but I wonder (perhaps naively) whether 'university' should have been included in the model as a clustering term, especially since some of the earlier literature cited indicates that the environment (beyond the mentor and lab) can contribute to RRP.

Thanks for flagging this, we agree and reran our models to see if university functioned as a potential confounding variable. This would mean that part of the association between behavior of the supervisor and the PhD candidate was actually due to the university environment. We now present crude and adjusted models side-by-side in Table 2. The upshot is that the institutional environment is of greater relevance for open access, but the associations identified for open data stand and are even strengthened by adding the environment.

DiscussionThe first paragraph of the Discussion contains information that I would ordinarily associate with Results. That aside, can you clarify the interpretation of the open-access information? If these are presented as exp(b) values, then would a.18 would indicate that PhD candidates published open access less often when their supervisors did than when they did not?

Apologies for the confusion, the reviewer is right. When looking into this, we found that the recoding of our variable (never open access/sometimes open access/often open access) was flawed, as recent data showed that 76% of publications from Dutch researchers are open access (https://www.rathenau.nl/en/science-figures/output/publications/open-access-research-publications). Hence the groups we created when recoding were heavily skewed. We updated the preregistration, detailing the rationale for choosing a different coding scheme and recoded Open Access into up to the national average (0) and beyond the national average now (1). This created a binary variable. We reasoned that supervisors who publish Open Access more often than the national average could be seen as actively practicing open access, and thus as role models. We now write:

“Effects for Open Access were smaller (1.99) and became nonsignificant when correcting for the role of the institution.”

Contextualization: Since there is a substantial cost associated with publishing in many open-access venues, might an important contributor to variance (that might affect this model but is not included in the model) be the funding amount of the supervisor? For example, some supervisors may have limited ability to publish open access where a cost is incurred, some may be able to publish their own work (or at least some of their own work) open access, but fewer likely can afford to support their candidates in publishing at cost. If there is indeed a weaker relationship between supervisor publishing and candidate publishing than between data availability, then could this be a mediating or moderating factor?

This could indeed be true, and we added this to the Contextualisation section (Discussion), while noting that green OA is still open, but free of (financial) cost:

“Open access comes in different forms and publishing in open access journals is often not for free. Some publishers make exceptions for scientists from low and middle income countries, but The Netherlands would rightly not classify. It could thus be that the amount of funding that a supervisor or PhD candidate had available affected the relationship we studied. In other words: funding availability may determine whether PhD candidates (or supervisors) chose to publish in an open access journal. However, it should be noted that green open access, archiving a paper in an appropriate format in an (institutional) repository, can be done free of financial charge.”

The role of ECRs: I'm not sure that we should assume that ECRs have more opportunities to learn about responsible research, in general, though they may have more exposure to open research principles. This paper primarily focuses on open data and open publication. While these are important components of reproducibility and transparency, and while they may aid in the identification of problematic findings and work, they do not subsume the entirety of RRPs. Some components of RRP are longstanding or axiomatic ethical determinations or orientations, whereas the rise of open access has been relatively recent. So even highly ethical, responsible, and transparent senior researchers may be slow to uptake new publishing approaches.

We now refer to open science practices, and tried to make it clear from the outset that these are, as you rightly indicate, only a subset of RRPs. It seems true that many activities to educate researchers about open science practices do focus on ECRs, hence we trust the paragraph to be in order, given the greater current emphasis on open science.

The role of ECRs: Since your findings are nondirectional (e.g., correlations), it seems plausible (given the citations you provide) that you may be capturing a bidirectional relationship.

True, we incorporated this so that the paragraph now describes the possibility of PhD candidates influencing supervisors, supervisors influencing PhD candidates, or a bidirectional relationship. We end the paragraph with the following sentence:

“Finally, it could be that the relationship investigated here is bidirectional.”

[Editors' note: further revisions were suggested prior to acceptance, as described below.]

Reviewer #2 (Recommendations for the authors):I would like to thank the authors for the work put into this revision and the clarifications offered in the response to reviewers. I restate my perception that this work is interesting and provides a solid and useful contribution to the overall scientific enterprise in this area. I have a few questions and comments around the revised work that I hope you find useful.AbstractHere and elsewhere, you might want to use or provide the specific meaning of "actively published open access" since it's very specific to this article (e.g., published open access more often than the average Dutch professor did in 2021). There are also comments in the Discussion about the odds ratio that apply to the abstract (depending on whether you decide to make those revisions).

Thank you for flagging this, we now use the more descriptive “more often than the national average” to prevent confusion.

IntroductionI think that the modifications made to the Introduction section are helpful and address my comments in full.

Thank you.

Materials and methodsI am glad that my questions prompted further investigation into the analyses and variable structure. My only remaining comment relates to the new statement, "The 10% rule for confounding was applied." I think the application here is fine (again – I am not a professional statistician, so caveat emptor) but a citation may be useful. Some readers may be unfamiliar with that rule, and others may have questions about it (see, eg., Lee, 2014 – https://doi.org/10.2188%2Fjea.JE20130062).

Many thanks for suggesting this, we have added some references to sources (Budtz-Jørgensen et al., 2007; Beukelman & Brunner, 2016) that provide an accessible introduction to the confounder rule applied.

ResultsIn the title for Table 1, you might want to specify the unit of analysis (e.g., a unique DOI). While it is clear to me what is meant by the table, a reader who is skimming might assume that there are, for example 548 PhD candidates publishing open access, rather than 548 DOIs from PhD candidates that were published open access.

We have adjusted the description of the table accordingly, thanks for the close reading. It now reads:

“Prevalence for each practice (expressed as unique DOIs) appears in Table 1, as well as correlations between the PhD candidates’ engagement in a practice and the supervisors’ engagement in a practice.”

The new title for Table 1 is:

“Prevalence of Open Access publishing and sharing data openly among unique DOIs.”

If I understand your methods for identifying open data correctly, it may be worth removing the "Open data (automated)" information from the Results entirely. Here is my reasoning: Oddpub was used to extract possible instances of open data statements, but through manual verification, you triaged around half of those instances. Specifically, it seems that those that were excluded by manual verification did not have open data statements as affirmed by two to three human reviewers. Thus, I am not sure what value is added by analyzing the "Open data (automated)" variable, which seems to only describe a subset of papers for which a specific algorithm thought it found open data. We are not concerned with whether things are associated with whether a machine thinks there might be an open data statement, but rather whether things are associated with actual open data access.

We understand your reasoning but respectfully disagree for a number of reasons. First, The Oddpub tool by Riedel and colleagues (2020) has fairly good sensitivity (0.73) and excellent specificity (0.97) and has been used around the world. Hence, we would prefer to keep the passages in as it allows for international comparison. In addition, keeping it in allows others to re-do our analyses for integrity purposes. Also, the automated screening is arguably a less resource intensive manner (as compared to human deliberation) to assess data sharing. Finally, the sample size and power was calculated based on automated Open Access screening, not on human deliberation, and hence we would prefer to present both results side-by-side as we believe that they provide the most accurate description together.

Can you please consider verifying the total number of linked publications (81,091)? Given that the supervisor+candidate total n was around 2,000, that number would imply that the supervisors in this study published an additional 79,000 or so papers before 2017. While that is certainly possible, even if we assume that the 211 recent PhD candidates published an aggregate 10,000 papers (a very generous assumption), it would mean that the 211 supervisors published an average of 327 papers each, which is fairly remarkable. In some ways, this number is less relevant than the 2 retractions, but it does make me wonder.

We verified this. A few contextual remarks may be helpful. Firstly, researchers may have published since we stopped sampling last year. Secondly, the number of unique papers is smaller, as many publications are multi-authored. Thirdly, it is not entirely helpful to speak of averages here as we found 20% of the authors to account for 64% of the publications (see excel sheet attached to this revision and added to the GitHub repository). Note that for the first researcher who started publishing in 1982, over 2000 publications were identified.

DiscussionIf you agree with my point about automated open data statement identification, you can also remove this sentence from the Discussion ("Based on the automated detection of data sharing statements, we found that having a supervisor that shared data is associated with 2.21 times the odds to share data when compared to having a supervisor that did not share data.")

As above, we would prefer to keep the passage in as the two ways of assessing data sharing together provide the most accurate description.

I would suggest revising this sentence: "Effects for Open Access were smaller (1.99) and became nonsignificant when correcting for the role of the institution." Instead, you could indicate that the unadjusted odds ratio was 1.99 (p=.011) and the adjusted odds ratio was 1.64 (p=.079). I think that the shift in odds is interesting even though it falls above the conventional significance threshold.

We agree this is a cleaner presentation of the main findings and now write:

“ The unadjusted open access odds ratio was 1.99 (p=.011) and became 1.64 (p=.079) when correcting for the role of the institution.”

You currently write, "This may indicate there being greater acceptance of and support for a particular practice in a research group. In other words: a responsible supervisory climate may empower PhD candidates to engage in open science more readily." You might want to contextualize this with something like, "In considering the associations that we identified, we speculate that…"

Thank you for suggesting this, we have largely taken over your formulation and now write:

“Considering the associations that we identified, we speculate that working under supervisors who engage in open science themselves could empower PhD candidates to engage in open science more readily.”

References

Riedel, N., Kip, M. and Bobrov, E., (2020). ODDPub – a Text-Mining Algorithm to Detect Data Sharing in Biomedical Publications. *Data Science Journal*, 19(1), 1-14. http://doi.org/10.5334/dsj-2020-042

Beukelman, T., & Brunner, H. I. (2016). Chapter 6 – Trial Design, Measurement, and Analysis of Clinical Investigations. In *Textbook of Pediatric Rheumatology* (7th ed., pp. 54–77). W.B. Saunders. https://doi.org/10.1016/b978-0-323-24145-8.00006-5

Budtz-Jørgensen, E., Keiding, N., Grandjean, P., & Weihe, P. (2007). Confounder selection in environmental epidemiology: assessment of health effects of prenatal mercury exposure. *Annals of epidemiology*, *17*(1), 27–35. https://doi.org/10.1016/j.annepidem.2006.05.007